# Fast *in-vitro* screening of FLT3-ITD inhibitors using silkworm-baculovirus protein expression system

**Naoki Yamamoto**[1]*, **Jiro Kikuchi**[2], **Yusuke Furukawa**[2], **Naoya Shibayama**[1]*

**1** Division of Biophysics, Department of Physiology, School of Medicine, Jichi Medical University, Shimotsuke, Tochigi, Japan, **2** Division of Stem Cell Regulation, Center for Molecular Medicine, School of Medicine, Jichi Medical University, Shimotsuke, Tochigi, Japan

* nyamamoto@jichi.ac.jp (NY); shibayam@jichi.ac.jp (NS)

## Abstract

We report expression and purification of a FLT3 protein with ITD mutation (FLT3-ITD) with a steady tyrosine kinase activity using a silkworm-baculovirus system, and its application as a fast screening system of tyrosine kinase inhibitors. The FLT3-ITD protein was expressed in *Bombyx mori* L. pupae infected by gene-modified nucleopolyhedrovirus, and was purified as an active state. We performed an inhibition assay using 17 kinase inhibitors, and succeeded in screening two inhibitors for FLT3-ITD. The result has paved the way for screening FLT3-ITD inhibitors in a fast and easy manner, and also for structural studies.

**Data Availability Statement:** All relevant data are within the paper and its Supporting information files.

## 1. Introduction

Acute myeloid leukemia (AML) is a proliferative malignancy of hematopoietic cells which are blocked in differentiation at a variety of stages [1]. Several mutations in proteins, such as tyrosine kinases, transcription factors, and epigenetic factors, are related to the onset of AML [2]. Among them, FLT3 (Fms-Like Tyrosine kinase-3) with internal tandem duplication (ITD) occurring at the juxtamembrane (JM) domain (Fig 1A, gray and yellow parts), which we term FLT3-ITD, is known to be a poor-prognostic factor [3,4]. The mutation results in constitutive activation of FLT3 tyrosine kinase due to ligand-independent homodimerization. Several drugs have been known to be effective to suppress the activity of FLT3-ITD [5,6]. For example, midostaurin (PKC412) is one of the potent drugs, and has clinically been used [7]. Recently, gilteritinib (ASP2215) has also been released to the market [8]. Yet discovery of other potent drugs is desired for overcoming resistance to existing drugs, and for broadening choices of the clinical treatment.

Drug screening for AML has been performed using cell-proliferation systems in which AML is stimulated upon mutation on related genes. The effect of drugs can be confirmed from the death of the cells. However, the span for cell cultivation required to confirm the effect of drugs usually takes more than two days [7,9–11], and also there still remains a possibility that the regression of the cell life is not directly caused by the suppression of the protein function stimulated by the mutation [12]. Therefore, direct evaluation of the effect of drugs on the target

**Funding:** The authors received no specific funding for this work.

**Competing interests:** The authors have declared that no competing interests exist.

protein is desirable for the more rapid and accurate drug screening. To date, there have been some commercially-available FLT3-ITD proteins by which the kinase activity can be evaluated [13]. However, since the purity of these proteins is not sufficiently high, well-purified FLT3-ITD proteins are desired for biochemical and especially for structural studies.

In this study, we report a construction of an expression system for a FLT3-ITD protein using silkworm pupae, and its adaptation to high-throughput drug screening system. The protein was able to be purified as a single component from the pupae as an active state. A drug screening using several candidates was performed, and potent drugs to FLT3-ITD were successfully identified. As far as we know, it is the first time to obtain active recombinant FLT3-ITD proteins as a single component.

## 2. Materials and methods

### 2.1. Recombinant baculovirus production for the expression of the FLT3-ITD protein

The FLT3-ITD protein was expressed in the Silkworm-Baculovirus System produced by Pro-Cube (Sysmex Corporation, Kobe, Japan). Briefly, a gene sequence corresponding to the tyrosine kinase domain with an ITD mutation (Fig 1A) was amplified via PCR using primers (forward: 5'-cgcggtaccATGCACAAGTACAAAAAGCAATTTA and reverse: 5'-ggctcta gaCACATTCTGATACATCGCTTCTTCTG where the lower-case characters indicate codons for the further ligation reactions and the capital ones part of the FLT3-ITD gene). The amplified oligo DNA was inserted in the pM23 transfer vector (Sysmex Corporation, Kobe, Japan) at

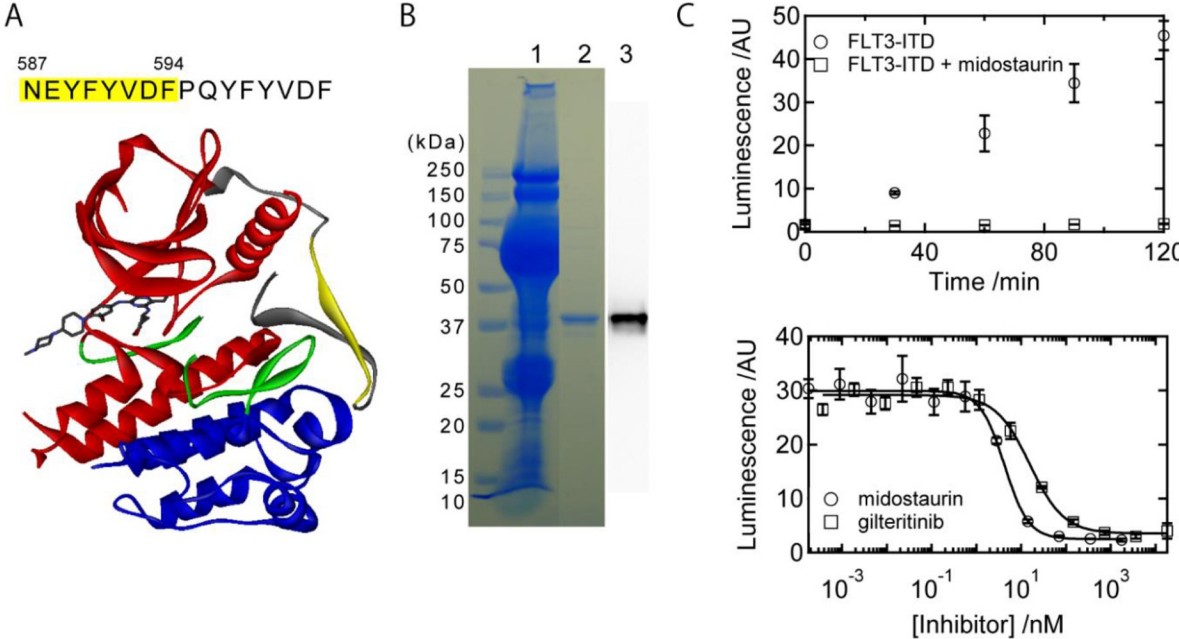

**Fig 1.** (A) The duplicated and template (indicated by yellow) amino acid sequence of the FLT3-ITD protein. Bottom is the crystal structure of the WT FLT3 protein in complex with giliteritinib (PDB ID; 6JQR). The gray, red, green, and blue parts represent the JM domain, N robe, activation loop, and C robe, respectively. The yellow part within the JM domain is the template sequence for the duplication. (B) SDS-PAGE of the purification process. Lane 1 is the supernatant of the ground pupae collected by ultracentrifuge. Lane 2 represents the purified protein by the ion-exchange chromatography, and the result of the western blot using anti-FLAG antibody is shown Lane 3. (C, top) Time dependent measurement of the tyrosine kinase activity of the purified FLT3-ITD protein. As a control, midostaurin was used to inhibit the activity. (C, bottom) The inhibitory curves of midostaurin and gilteritinib. The minimal data sets used for these figures are represented in Supporting Information.

KpnI and XbaI cites. The pM23 vector is designed to conjugate the FLAG tag sequence to the C-terminal of the recombinant protein. This gene-transferred pM23 vector was co-transfected with linearized *B. mori* nucleopolyhedrovirus (i.e. BmNPV) DNA [14] into a *B. mori*-derived cell line (i.e. BmN) [15]. If the homologous recombination between these two DNA successfully occurs, the recombinant virus obtains the replication function. After 7 days of incubation at 25˚C, the recombinant virus existing in the supernatant was injected into the body of silkworm pupae. 50 µl of the virus at $10^5 \sim 10^6$ PFU/ml was used for the infection. The infected pupae were harvested for 6 days at 25 ºC and 50% humidity, and frozen.

## 2.2. Purification of the FLT3-ITD protein

The pupae infected by the baculovirus were homogenized with a buffer solution (20 mM Tris-HCl, 150 mM NaCl, 5 mM DTT, 1mg/ml phenylthiourea, one inhibitor cocktail tablet (*EASYpack*, Roche, Basel, Switzerland) per 50 ml solution, pH 8.0). The homogenized solution was centrifuged to separate the supernatant and precipitate using an ultracentrifuge (100,000 g) for 1 h at 4˚C. The supernatant was filtered using 0.8 µm Minisart Syringe Filter (Sartorius, Göttingen, Germany), and purified using DDDDK-tagged protein purification gel (MBL, Japan) followed by elution with 0.1 mg/ml FLAG Peptide (Sigma-Aldrich Co., St. Louis, Missouri, the US). To remove the flag peptide, further purification by the ion-exchange chromatography was performed using ToyoScreen SuperQ-650M column (TOSOH, Yamaguchi, Japan) built up with AKTA purifier (GE healthcare, Chicago, Illinois, the US). The purified FLT3-ITD protein was used for the kinase activity measurements. The concentrations of the protein used for the kinase activity measurement were 0.2–0.4 µM which were estimated by Western Blot using an anti-FLAG antibody, Anti-DDDDK-tag mAb-HRP-Direct (MBL, Japan).

## 2.3. Kinase activity measurement

The kinase activity of the FLT3-ITD protein was measured by using ADP-Glo™ Kinase Assay Kit (Promega, Madison, Wisconsin, the US). Briefly, the protein solution was incubated in 0.2 mM DTT, 0.1 mg/ml myelin basic protein, 0.1 mM ATP, the reaction buffer, and 10% DMSO. The drug concentration at 1 µM was used for monitoring the inhibition activities. The kinase reaction was initiated upon the addition of ATP, and 15 µl of the reaction solution was mixed with the same amount of ADP-Glo Solution. 40 min after the incubation, 30 µl of Kinase Detection Reagent was added to the solution. 30 min after the incubation, the luminescence intensity was monitored using SPARK 10M (TECAN, Zurich, Switterland) in a 96-wel plate. The measurement was replicated three times and the standard deviation at 95% confidence interval was calculated. To draw the inhibitory curves, the protein solution in the presence of inhibitors at various concentrations was incubated for 2 h and the same detection protocol was performed.

## 3. Results and discussion

### 3.1. Purification of the FLT3-ITD protein

In this study, we chose a FLT3-ITD protein found in a clinical case (Fig 1A) [16]. The full amino-acid sequence expressed in this study is shown in Supporting Information. PQYFYVDF is the duplicated sequence where NEYFYVDF prior to the sequence was supposed to be the template of the duplication (indicated by the yellow highlight). The purification by the FLAG tag already yielded an almost pure protein solution. The ion-exchange chromatography was performed to exclude the FLAG peptide, and it was confirmed that the purified FLT3-ITD protein appeared as a single band on SDS-PAGE stained with Coomassie brilliant blue

(Fig 1B). The proper expression of the protein was also confirmed by Western Blot using anti-FLAG antibody (Fig 1B). The protein yield was ~20 μg per one pupa.

## 3.2. The kinase activity of the expressed FLT3-ITD protein

Time-dependence of the kinase activity of the FLT3-ITD protein is shown in Fig 1C top. The kinase activity increases in a time-dependent manner. On the contrary, no such increase was observed in the presence of the inhibitor, midostaurin. The results represent that the expressed protein possesses a steady kinase activity, which is detectable by the standard assay method. The absolute protein activity was calculated using a standard calibration method, and the ATP-to-ADP conversion rate was obtained to be ~50 nmol/min/mg.

The inhibition curves by midostaurin and gilteritinib are shown in Fig 1C bottom. These two inhibitors certainly inhibited the kinase activity in the concentration-dependent manner. The inhibition curves were fitted by the hill function to obtain the IC50 values. As a result, the IC50 values were 4.2±0.1 nM and 16±3 nM (S.D. obtained by the curve fitting) for midostaurin

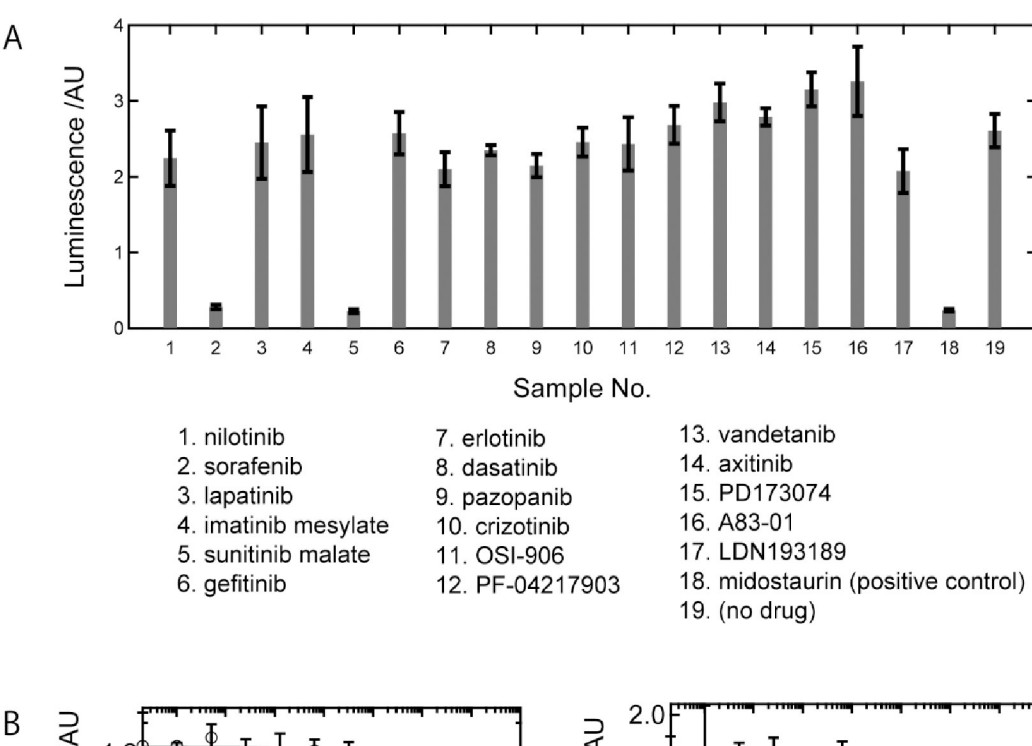

1. nilotinib
2. sorafenib
3. lapatinib
4. imatinib mesylate
5. sunitinib malate
6. gefitinib
7. erlotinib
8. dasatinib
9. pazopanib
10. crizotinib
11. OSI-906
12. PF-04217903
13. vandetanib
14. axitinib
15. PD173074
16. A83-01
17. LDN193189
18. midostaurin (positive control)
19. (no drug)

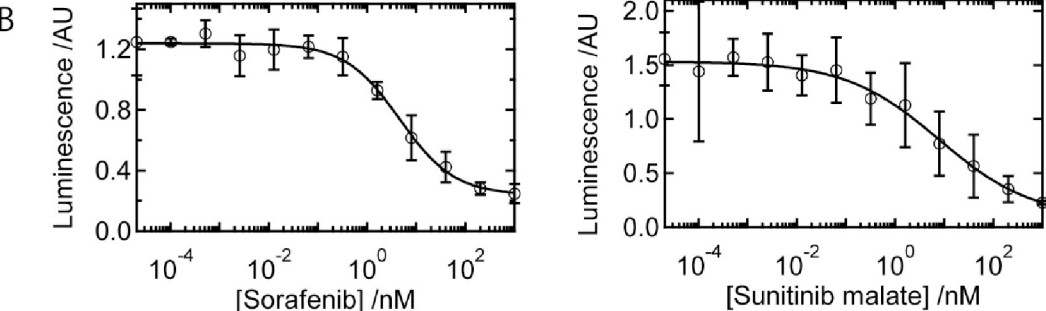

**Fig 2.** (A) Inhibition of the tyrosine kinase activity of the FLT3-ITD protein by various drugs. Sorafenib and sunitinib possess the same activity as that of midostaurin. (B) The inhibitory curves of sorafenib and sunitinib. The minimal data sets used for these figures are represented in Supporting Information.

and gilteritinib, respectively. The value of midostaurin is two-order lower than that reported in *in-vivo* study [7], indicating that our screening system possesses a high sensitivity.

### 3.3. Screening assay of various potentially-useful drugs against the expressed FLT3-ITD protein

The inhibition activity of various kinds of kinase inhibitors was monitored. We tried 17 kinase inhibitors that have already known to inhibit kinase activities. Fig 2A shows the inhibitory activities of the compounds. As controls, midostaurin (positive control) and buffer (negative control) were also tested. Sorafenib and sunitinib, which have been known to be effective to FLT3-ITD mutated cell lines [17], had clear inhibitory effects whereas the other drugs did not show apparent activities. The inhibitory curves of the two drugs were obtained and fitted by using the hill function as shown in Fig 2B. As a result, the IC50 values were 4.6±0.9 nM and 8.2±4.3 nM (S.D. obtained by curve fitting) for sorafenib and sunitinib, respectively. These values are as low as those of midostaurin, confirming that these two drugs are potent inhibitors of the FLT3-ITD protein.

## 4. Conclusion

As far as we know, it is the first time to successfully express and purify soluble FLT3-ITD protein. Thus, the present expression system using the silkworm pupae is a quite effective way to obtain the active FLT3-ITD protein.

In the screening experiment, sorafenib and sunitinib were successfully found to be effective to inhibit the tyrosine kinase activity of the FLT3-ITD protein. The screening is fast (within 2 hours) and ~1,000 trials can be done per one pupa. Therefore, the expression system built in this study is a powerful method to find new tyrosine kinase inhibitors. Furthermore, the protein obtained as a highly-purified state is suitable for biochemical and especially for structural determination studies.

## Supporting information

**S1 File.**
(DOCX)

**S1 Raw images.**
(PDF)

## Author Contributions

**Conceptualization:** Yusuke Furukawa, Naoya Shibayama.

**Data curation:** Naoki Yamamoto.

**Formal analysis:** Naoki Yamamoto.

**Investigation:** Naoki Yamamoto, Jiro Kikuchi, Yusuke Furukawa, Naoya Shibayama.

**Project administration:** Naoya Shibayama.

**Resources:** Naoki Yamamoto, Jiro Kikuchi, Yusuke Furukawa, Naoya Shibayama.

**Supervision:** Naoki Yamamoto.

**Writing – original draft:** Naoki Yamamoto.

**Writing – review & editing:** Naoki Yamamoto, Yusuke Furukawa, Naoya Shibayama.

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
