## [Decision Letter · Decision Letter 0]

16 Feb 2022

PONE-D-21-38649Fast in-vitro screening of FLT3-ITD inhibitors using silkworm-baculovirus protein expression system

PLOS ONE

Dear Dr. Yamamoto,

Thank you for submitting your manuscript to PLOS ONE. After careful consideration, we feel that it has merit but does not fully meet PLOS ONE’s publication criteria as it currently stands. Therefore, we invite you to submit a revised version of the manuscript that addresses the points raised during the review process by Reviewer #2.

We look forward to receiving your revised manuscript.

Kind regards,

Francesco Bertolini, MD, PhD

Academic Editor

PLOS ONE

Journal Requirements:

Reviewers' comments:

Reviewer's Responses to Questions

**Comments to the Author**

1. Is the manuscript technically sound, and do the data support the conclusions?

Reviewer #1: Yes

Reviewer #2: Partly

2. Has the statistical analysis been performed appropriately and rigorously? 

Reviewer #1: Yes

Reviewer #2: Yes

3. Have the authors made all data underlying the findings in their manuscript fully available?

Reviewer #1: Yes

Reviewer #2: Yes

4. Is the manuscript presented in an intelligible fashion and written in standard English?

Reviewer #1: Yes

Reviewer #2: Yes

5. Review Comments to the Author

Reviewer #1: In the era of personalized medicine, it becomes important to have effective and fast techniques for the detection of active molecules. This work clearly and rigorously presents a useful and fast system for this type of activity. The point that could be developed in the future is the study of the different lengths of the ITD on the efficacy of the inhibition.

Reviewer #2: Major comments：

1. To date, there have been some commercially-available FLT3-ITD proteins by which the kinase activity can be evaluated. Why are you still doing this work? please provide evidence if you purified protein better than the commercial one.

2. In the abstract， “We performed an inhibition assay using 17 potential kinase inhibitors, and succeeded in identifying two potent inhibitors for FLT3-ITD.” It is easy to mislead readers into thinking that you have found two new inhibitors.

Minor comments：

Some details of the materials and methods need to be supplemented, the following are specific examples:

1. What is the template for ITD mutation sequence PCR?

2. When co-transfected pM23 vector with linearized BmNPV DNA, what is the transfection reagent and what is the amount of transfection system? And what are the cell culture conditions?

3. Obviously, not all of the homologous recombination between the two DNA can successfully occurs, how did you screen that?

4. What is amount of the recombinant virus used for injection? And after injection, what conditions are the pupae placed in?

Next are some of my questions or suggestions:

1. During the 7 days incubation period after transfection, I would like to know if you change the medium for the old cells or collect the recombinant virus to re-infect the new cells?

2. Fig 1. (B) Lane 1 and Lane 2 appear to be two images stitched together; and I think adding one lane, the supernatant of the normal pupae (blank control) would be better.

3. In the Results and discussion, 3.1, SDS PAGE missed a “-”.

4. In conclusion, “Thus, the present expression system using the silkworm pupae is quite effective way to obtain the active FLT3-ITD protein”, before “quite” need to add “a”.

5. The references may be a little less, and mostly not up to date.

6. PLOS authors have the option to publish the peer review history of their article (what does this mean?). If published, this will include your full peer review and any attached files.

Reviewer #1: No

Reviewer #2: No

---

## [Author Response · Author response to Decision Letter 0]

3 Mar 2022

Response to the reviewers’ comments

(The yellow highlighted parts are those modified in the main text)

Reviewer #1: In the era of personalized medicine, it becomes important to have effective and fast techniques for the detection of active molecules. This work clearly and rigorously presents a useful and fast system for this type of activity. The point that could be developed in the future is the study of the different lengths of the ITD on the efficacy of the inhibition.

Reply

We appreciate that the importance of our study is well understood. As the reviewer mentioned, we would like to deal with FLT-ITD proteins with different ITD lengths, so that new drugs optimized for the personalized medicine will be efficiently found.

Reviewer #2: Major comments：

1. To date, there have been some commercially-available FLT3-ITD proteins by which the kinase activity can be evaluated. Why are you still doing this work? please provide evidence if you purified protein better than the commercial one.

Reply

Thank you for giving a comment discussing about the importance of our study. We would like to answer them by the point-by-point manner.

As the reviewer mentioned, there has already been commercially-available FLT3-ITDs. For example, SignalChem supplies a FLT3-ITD protein (https://signalchem.com/product_details.php?id=ZWE1ZDJmMWM0NjA4MjMyZTA3ZDNhYTNkOTk4ZTUxMzUtMTE5MDM=#specifications). However, the purity of the protein is low because it contains non-negligible contaminated proteins around 26 kDa (Figure R1). On the other hand, in our case the FLT3-ITD protein appears as the major band in the SDS-PAGE as shown in Figure 1B, which represents that we purified protein better than the commercial case.

2. In the abstract， “We performed an inhibition assay using 17 potential kinase inhibitors, and succeeded in identifying two potent inhibitors for FLT3-ITD.” It is easy to mislead readers into thinking that you have found two new inhibitors.

Reply

Following what the reviewer suggested, we have changed the corresponding sentence as follows;

(Page 2, line 6)

We performed an inhibition assay using 17 kinase inhibitors, and succeeded in screening two inhibitors for FLT3-ITD.

In addition to this, one sentence in conclusion was also modified as follows;

(page 10, line 3)

As far as we know, it is the first time to successfully express and purify soluble FLT3-ITD protein.

(Original sentence; As far as we know, it is the first time to successfully express a soluble FLT3-ITD protein.)

Minor comments：

Some details of the materials and methods need to be supplemented, the following are specific examples:

1. What is the template for ITD mutation sequence PCR?

Reply

We added the DNA sequence confirmed in the pM23 vector in Supporting Information.

2. When co-transfected pM23 vector with linearized BmNPV DNA, what is the transfection reagent and what is the amount of transfection system? And what are the cell culture conditions?

Reply

The transfection reagent and its amount are not open because such information is protected under the patent of ProCube system owned by Sysmex Corporation. The cell culture was harvested at 25 ºC. We added this information in Materials and Methods.

3. Obviously, not all of the homologous recombination between the two DNA can successfully occurs, how did you screen that?

We did not confirm if each cell was infected. However, we were able to tell that 80-90 % of the cells were infected by the virus judging from their appearance.

4. What is amount of the recombinant virus used for injection? And after injection, what conditions are the pupae placed in?

Reply

50 μl of the virus at 105～106 PFU/ml was used for the infection. The infected pupae were put at 25 ºC and 50 % humidity. We added these experimental conditions in Materials and Methods.

Next are some of my questions or suggestions:

1. During the 7 days incubation period after transfection, I would like to know if you change the medium for the old cells or collect the recombinant virus to re-infect the new cells?

Reply

We did not change the medium.

2. Fig 1. (B) Lane 1 and Lane 2 appear to be two images stitched together; and I think adding one lane, the supernatant of the normal pupae (blank control) would be better.

Reply

Since we do not find the merit to add the supernatant of the normal pupae to the SDS-PAGE result, we do not include it.

3. In the Results and discussion, 3.1, SDS PAGE missed a “-”.

Reply

We added the missing hyphen at the corresponding place.

4. In conclusion, “Thus, the present expression system using the silkworm pupae is quite effective way to obtain the active FLT3-ITD protein”, before “quite” need to add “a”.

Reply

We added the missing word at the corresponding place.

5. The references may be a little less, and mostly not up to date.

Reply

We added some up-to-date references in Introduction.

---

## [Decision Letter · Decision Letter 1]

30 Mar 2022

Fast in-vitro screening of FLT3-ITD inhibitors using silkworm-baculovirus protein expression system

PONE-D-21-38649R1

Dear Dr. Yamamoto,

Sorry for the delay. One of the two original reviewers was no longer available so the process was longer than usual as I had to check that all her/his requests were fullfilled.

We’re pleased to inform you that your manuscript has been judged scientifically suitable for publication and will be formally accepted for publication once it meets all outstanding technical requirements.

Kind regards,

Francesco Bertolini, MD, PhD

Academic Editor

PLOS ONE

Additional Editor Comments (optional):

Reviewers' comments:

Reviewer's Responses to Questions

**Comments to the Author**

1. If the authors have adequately addressed your comments raised in a previous round of review and you feel that this manuscript is now acceptable for publication, you may indicate that here to bypass the “Comments to the Author” section, enter your conflict of interest statement in the “Confidential to Editor” section, and submit your "Accept" recommendation.

Reviewer #1: All comments have been addressed

2. Is the manuscript technically sound, and do the data support the conclusions?

Reviewer #1: Yes

3. Has the statistical analysis been performed appropriately and rigorously? 

Reviewer #1: Yes

4. Have the authors made all data underlying the findings in their manuscript fully available?

Reviewer #1: Yes

5. Is the manuscript presented in an intelligible fashion and written in standard English?

Reviewer #1: Yes

6. Review Comments to the Author

Reviewer #1: All the issue have been addressed in rigorous manner. There's no other question to answer. I hope that this system can be useful in future research.

7. PLOS authors have the option to publish the peer review history of their article (what does this mean?). If published, this will include your full peer review and any attached files.

Reviewer #1: No

---

## [Editor Report · Acceptance letter]

27 Apr 2022

PONE-D-21-38649R1 

Fast *in-vitro* screening of FLT3-ITD inhibitors using silkworm-baculovirus protein expression system 

Dear Dr. Yamamoto:

I'm pleased to inform you that your manuscript has been deemed suitable for publication in PLOS ONE. Congratulations! Your manuscript is now with our production department. 

Kind regards, 

on behalf of

Dr. Francesco Bertolini 

Academic Editor

PLOS ONE